# CCEval: A Representative Evaluation Benchmark for the Chinese-centric Multilingual Machine Translation

**Lianzhang Lou[1][§], Xi Yin[2][§][\*], Yutao Xie[2], Yang Xiang[1],**
[1]Peng Cheng Laboratory, Shenzhen, China
[2]International Digital Economy Academy, Shenzhen, China
`lzlou1993@gmail.com`,\*`yxthu@qq.com`,`yutaoxie@idea.edu.cn`,`xiangy@pcl.ac.cn`

## Abstract

The Chinese-centric Multilingual Machine Translation (MMT) has gained more importance recently due to increasing demands from international business development and cross-cultural exchanges. However, an important factor that limits the progress of this area is the lack of highly representative and high-quality evaluation benchmarks. To fill this gap, we propose *CCEval*, an impartial and representative Chinese-centric MMT evaluation dataset. This benchmark dataset consists of 2500 Chinese sentences we meticulously selected and processed, and covers more diverse linguistic features as compared to other MMT evaluation benchmarks. These sentences have been translated into 11 languages of various resource levels by professional translators via a rigorously controlled process pipeline to ensure their high quality. We conduct experiments to demonstrate our sampling methodology's effectiveness in constructing evaluation datasets strongly correlated with human evaluations. The resulting dataset enables better assessments of the Chinese-centric MMT quality. Our *CCEval* benchmark dataset is available at `https://bright.pcl.ac.cn/en/offlineTasks`.

## 1 Introduction

Multilingual Machine Translation (MMT) is a hot research area that has made big progress recently (Lin et al., 2020; Fan et al., 2021; Costa-jussà et al., 2022). As the language spoken by the largest population in the world, Chinese has become more and more influential globally, and China has been growing its worldwide impact. For example, in 2021, China's total import and export trade amount has reached 21% of the entire global trade amount[‡], which is a significant proportion. Despite this, the Chinese-centric MMT research has received less attention compared to the MMT study centered on English (Aharoni et al., 2019; Zhang et al., 2020a).

A critical challenge hindering the progress of the Chinese-centric MMT is evaluation, mainly due to the lack of representative and human-correlated evaluation datasets. Recently, a few evaluation benchmarks for MMT have emerged, such as FLORES-101 (Goyal et al., 2022) and Tatoeba (Tiedemann, 2020). However, they can only be used for English-centric translation evaluations (Goyal et al., 2022; Thu et al., 2016), have not undergone a rigorous human expert annotation and quality inspection process to ensure the dataset's high quality (Tiedemann, 2020), or are only limited to specific domains (Anastasopoulos et al., 2020). Furthermore, an ideal MMT evaluation dataset shall provide system-level evaluations that correlate closely with human assessments. To achieve a high correlation, Zhan et al. (Zhan et al., 2021) selected samples yielding substantial performance fluctuations of evaluated systems, while Liu et al. (Liu et al., 2022) allocated greater weights to samples with elevated translation chunk entropy. The restrictions of these works are their dependence on hypothesis text in the source sentence sampling process.

To overcome previous limitations as mentioned above, we propose a highly representative evaluation benchmark dataset, named *CCEval*, which is specifically constructed to enable evaluation of the Chinese-centric MMT models/systems. Each language pair in the *CCEval* contains 2,500 Chinese source sentences we carefully selected and processed using a diversity and translation difficulty based approach. Utilizing human-annotated Multidimensional Quality Metrics (MQM) (Freitag et al., 2021a) corpus from recent Workshop on Machine Translation (WMT) evaluation campaigns, we conduct experiments to demonstrate the validity of this approach in building evaluation datasets correlated better with human assessments. These

---

[§]Equal contribution.
[\*]Corresponding author.
[‡]https://oec.world/en/profile/country/chn

well selected Chinese source sentences have been professionally translated into eleven languages of various resource levels, i.e., Mongolian, Lao, Portuguese, Polish, Czech, Turkish, Italian, Russian, Arabic, Greek, and Croatian. Note that about half of them are low-resource languages. We employ a systematic annotation and translation quality inspection workflow to guarantee the dataset's high quality.

The *CCEval* dataset is accessible via BRIGHT[*], a Chinese-centric MMT evaluation platform. To foster relevant research using our benchmark, we validate the effectiveness of *CCEval* by conducting evaluation tests on actual MMT baseline models.

The contributions of our work are:

- We establish and publicly release a representative and high-quality MMT evaluation dataset. To the best of our knowledge, this is the first evaluation benchmark for the Chinese-centric MMT research community.

- We propose a Chinese source sentence selection criteria and sampling strategy based on the linguistic diversity and translation difficulty, facilitating the dataset's higher evaluation correlations with human assessments.

## 2 Dataset Construction

The overall construction process of our *CCEval* dataset includes three steps. In the first stage of source text screening, we consider various linguistic features of the Chinese sentences, and screen appropriate sentences from the initial text pool to satisfy the corresponding source diversity requirements. Subsequently, we propose a method to calculate the translation difficulty of each sentence, and further select highly representative samples based on this method. Finally, in the target sentence translation phase, we adopt a strict translation quality inspection pipeline to obtain a high-quality dataset.

### 2.1 Source text selection

In the source text screening stage, we first filtered sentences with sensitive words and meaningless text from the initial text pool. Then we focus on the linguistic diversity and the representativeness of the source sentences. The former requirement is important for the authority and fairness of the dataset, while the latter is essential to enhance the

correlation between dataset evaluation results and human judgments.

**Diversity-based selection** In terms of the linguistic diversity of the Chinese source sentences, we focus on the distributions of topic domain, word frequency, and grammar complexity. Our selection strategy is as follows:

1) We picked sentences from multiple domains and topics, evenly covering six major domains: News, Politics, Travel, Biomedical, Technology, and Daily Life, as well as 60 topics, as shown in the Appendix 1.1.

2) We obtain the appearing frequency of each candidate sentence according to a word-level frequency accumulation using a unigram language model. Our sampling distribution is similar to the obtained frequency distribution of the candidate sentences. The details are dictated in Appendix 1.3.

3) As an indicator of grammar complexity, we conducted syntactic analysis on the candidate sentences. Based on the results, we extracted sentences obeying an approximately normal distribution of the syntax tree depth. The details are also described in Appendix 1.3.

**Representativeness-based selection** Representativeness of the evaluation dataset means the degree of correlation with human assessments for the system ranking evaluations. A test set with more consistent evaluation results with human judgments is considered to have higher representativeness, and vice versa.

In this stage, we selected samples with the top 20% translation difficulty to improve the representativeness of the dataset. We verify the validity of this difficulty-based strategy in picking high-representativeness samples by performing experiments, as shown in Sec. 3.1.

In light of the sentence translation characteristics disclosable by back translations, we borrow the idea of denoising autoencoder (DAE) to characterize the degree of translation difficulty of a sentence. Specifically, we employ a DAE to perform cross-lingual corruption of the source sentence, and then use its degree of recovery to measure the translation difficulty of the sample. The formula for calculating the translation difficulty $d$ of a Chinese sentence is as follows:

$$d = 1 - s(x, x_r) \qquad (1)$$

where $x$ is the representation of the source sentence, and $x_r$ is the representation of the restored

[*]https://bright.pcl.ac.cn/en/

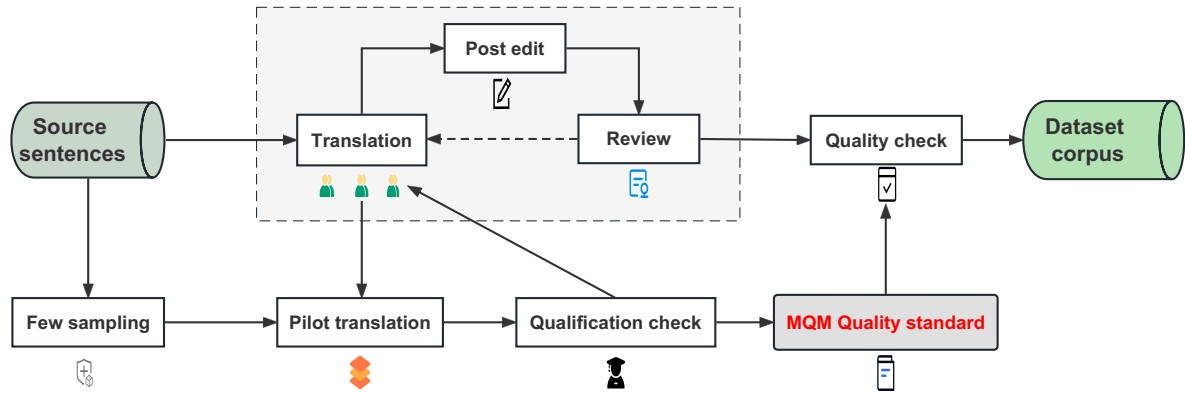

Figure 1: **Workflow of Translation Quality Control in *CCEval*.**

| Language | lo | mn | pt | pl | cs | ru |
|---|---|---|---|---|---|---|
| **Quality** | 91.8% | 90.7% | 91.1% | 92.7% | 93.3% | 95.0% |
| **Language** | it | tr | ar | el | hr | |
| **Quality** | 91.8% | 90.7% | 91.1% | 92.7% | 95.0% | |

Table 1: **Translation Quality Score of All Languages.** lo, mn, pt, pl, cs, ru, it, tr, ar, el, and hr denote Lao, Mongolian, Portuguese, Polish, Czech, Russian, Italian, Turkish, Arabic, Greek, and Croatian, respectively.

source sentence. The similarity metric $s$ is computed using the BERT-score Recall (Zhang et al., 2020b). Since we use this approach to further select source sentences for each language pair involved in *CCEval*, in the DAE process we utilized open-sourced corpora to train two separate Chinese2Many and Many2Chinese MMT models.

## 2.2 Translation annotation and quality management

Quality of the annotated translation from the Chinese source text to other languages is crucial to ensure the fairness of MMT evaluations. Therefore, strict translator qualification review and translation quality control process are applied to guarantee the translation quality of *CCEval*.

The whole workflow of translation annotation and quality control is shown in Fig. 1.

In the source text translation phase, we follow the translation annotation workflow of FLORES-101 (Goyal et al., 2022). For the selected Chinese sentences, we have engaged 11 different Language Service Providers (LSPs) to annotate the translations in their respective languages. Detailed information about the annotators of different languages is shown in Appendix 1.6.

After that, our linguistic experts implement the human quality inspection process. These professional linguistic experts are well qualified and have

| #Sentences | 2500 |
|---|---|
| Average Characters per Sentence | 30 |
| #Topics | 60 |
| % of Sentences with URL | 85% |

| Domain | # Sentences |
|---|---|
| | 2500 |
| Technology | 416 |
| Biomedical | 416 |
| Travel | 416 |
| Daily Life | 416 |
| News | 418 |
| Politics | 418 |

Table 2: **Statistics of *CCEval* Source Text.**

generally lived for 5 years or more in the target language-speaking countries or regions. They check the quality of each translated sentence provided by the LSPs based on Direct Assessments (DA). The translation quality is evaluated using a 5-point scale according to the fidelity and fluency, as detailed in Appendix 1.2, where translations receiving a score of 3 or greater are deemed correct.

The final translation quality scores of all eleven target languages are shown in Table 1. It is observed that all languages have a quality score above 90%, with a maximum of 95%.

## 2.3 Dataset overview

The following is a detailed breakdown of *CCEval*'s composition from multiple perspectives.

**Sources** Table 2 provides an overview of the *CCEval* source text. It includes 2,500 diverse and representative sentences. The mean length of our selected sentences is set to a moderate value of 30 characters, with long sentences exceeding 60 characters filtered. All samples contain rich metadata

| Dataset | #SPs | #Langs | CC | Div | Domain | #Topics | Quality | Source |
|---|---|---|---|---|---|---|---|---|
| FLORES-101 (Goyal et al., 2022) | 3001 | 101 | ✗ | ✗ | - | 10 | >90% | WikiNews/WikiJunior/WikiVoyage |
| ALT (Thu et al., 2016) | 20000 | 13 | ✗ | ✗ | News | - | - | WikiNews |
| TICO-19 (Anastasopoulos et al., 2020) | 3071 | 36 | ✗ | ✗ | Biomedical | 1 | >90% | PubMed/WikiNews et.al |
| CCEval | 2500 | 12 | ✔ | ✔ | 6 | 60 | >90% | Chinese Webs/Books |

Table 3: **Comparison of *CCEval* with Other MMT Evaluation Benchmarks**. **#SPs**, **#Langs**, **CC**, and **Div** denote the number of Sentence Pairs, the number of Languages, Chinese-centric, and Diverse, respectively. To the best of our knowledge, *CCEval* is the first Chinese-centric multilingual machine translation test set, which shows a diverse distribution in domain, topic, and other dimensions, as well as an equally tight control in dataset quality.

information, such as domain, topic, URL (except for the Daily Life domain), and created date and time, to facilitate source text tracing and in-depth research. This traceability is enabled by our selection of mainstream Chinese websites (detailed in Appendix 1.4) as the initial corpus pool of the source sentences.

**Domains** The dataset covers 6 representative domains and 60 distinct topics, as mentioned in Sec. 2.1. These domains are widely recognized, with News, Technology, and Dailylife domains adopted in WMT evaluations, Travel domain used in FLORES-101 (Goyal et al., 2022), and Biomedical domain specifically focused on by TICO-19 (Anastasopoulos et al., 2020).

**Languages** Our *CCEval* contains sentence pairs between Chinese and eleven languages (Table 1) of different families, including Slavic languages, Arabic languages, and Southeast Asian languages, with different levels of resources available. About half of these languages are low-resource languages, and they exhibit significant differences from Chinese in terms of morphology and grammar structure.

In Appendix 1.5, we provide some data examples of *CCEval*.

Table 3 shows the comparison of *CCEval* with other available MMT evaluation benchmarks. Overall, *CCEval* is the first Chinese-centric MMT test set with high quality. Compared to other mainstream benchmarks, our dataset has a comparable size and shows more diverse distributions in regard to domain, topic, and other linguistic features.

In particular, as shown in Fig. 2, we compare a typical source sentence distribution in *CCEval* with FLORES-101, in terms of sentence length, word frequency, and grammar complexity. It is observed that the source sentences in *CCEval* have approximately normal distributions of word frequency and grammar complexity. In contrast, their distributions in FLORES-101 are unbalanced and biased to the low ranges.

## 3 Experiments and Results

### 3.1 Empirical Verification: Validity of the difficulty-based strategy

We experimentally verify the effectiveness of using difficulty-based approach in selecting high- representativeness samples. Ideally the verification should have been done on the *CCEval* dataset directly, but it is very costly to do MQM/DA annotation of different systems' performance on *CCEval*, since this must be done by senior linguistic experts mastering both Chinese and our eleven languages. Alternatively, we validate the hypothesis by performing experiments on two publicly available WMT datasets with rich MQM annotation data, as well as an in-house test set with DA annotations of different systems.

Specifically, we first use Eq. 1 to calculate the sample difficulty from the two open-sourced Chinese-to-English translation evaluation sets of WMT 2021 and WMT 2022, and one in-house Chinese-to-Mongolian translation test set. Then we obtain the statistical difference between two sampling strategies using different subsets with varied difficulty levels, in terms of system ranking correlation with human assessments.

Figure. 3 shows the three translation evaluation datasets' system ranking correlations with the MQM/DA human evaluations under different subset selection strategies. For all three datasets, our proposed difficulty-based subset sampling strategy yields a higher correlation than random subset sampling. In addition, the selection of more difficult samples further enhances the correlations.

### 3.2 Evaluation of Multilingual Baseline Models using *CCEval* Benchmark

**Baselines** a) M2M-100 (Fan et al., 2021) is the first many-to-many machine translation model developed by Meta, which was trained on CCAligned corpus (El-Kishky et al., 2020) and is capable of arbitrary bilingual translations within more than

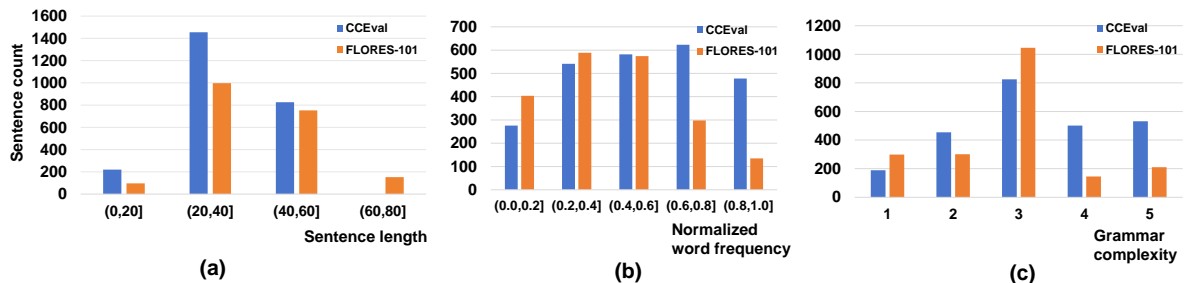

Figure 2: **Comparisons of a typical source sentence distribution in *CCEval* with FLORES-101.** *CCEval* possesses better linguistic diversity in terms of sentence length, word frequency, and grammar complexity.

| Resource level | | Low | | Medium | | | | | | | High | |
|---|---|---|---|---|---|---|---|---|---|---|---|---|
| **Language** | | lo | mn | tr | el | hr | pl | cs | pt | it | ar | ru |
| **M2M-100** | **from Chinese** | 0.69 | 0.81 | 10.40 | 11.75 | 9.76 | 8.87 | 8.91 | 12.95 | 14.09 | 7.45 | 11.38 |
| (Fan et al., 2021) | **to Chinese** | 3.89 | 5.09 | 14.09 | 12.19 | 12.73 | 11.95 | 12.36 | 12.61 | 13.02 | 12.33 | 13.43 |
| **mBART50** | **from Chinese** | - | 0.32 | 1.65 | - | 0.85 | 1.39 | 2.12 | 2.70 | 1.80 | 0.66 | 1.87 |
| (Tang et al., 2020) | **to Chinese** | - | 4.42 | 0.87 | - | 4.03 | 1.82 | 1.92 | 3.27 | 1.01 | 3.58 | 2.08 |

Table 4: **Multilingual translation evaluation results (SacreBLEU) using *CCEval* benchmark.**

100 languages. It covers all 11 languages in *CCEval*. b) mBART50 (Tang et al., 2020) is another multilingual translation model obtained by pre-training mBART (Liu et al., 2020) on 25 more languages and further finetuning on multilingual corpus, which covers 9 languages in *CCEval* except for Lao and Greek.

**Metric** We use SacreBLEU (Post, 2018) as the evaluation metric. For Chinese, we perform full-width to half-width conversion.

**Results** The evaluation results are shown in Table 4. The main findings are: Firstly, for both models and most languages, the SacreBLEU from Chinese to other languages is lower than that of the reverse direction, i.e., from other languages to Chinese. Furthermore, for the M2M-100 model, the SacreBLEUs between low-resource languages and Chinese are much lower than those between high- or medium-resource languages and Chinese.

# 4 Conclusion

In this paper, we construct and publicly release *CCEval*, a Chinese-centric MMT evaluation benchmark dataset covering twelve languages of varying resource levels, about half of which are low-resource languages. The dataset is highly representative, and the linguistic diversity of Chinese outperforms existing MMT benchmarks. We hope that the open source of this dataset will benefit the community. In the future, we will continue to expand the number of languages covered in *CCEval* and increase the dataset's sample size. We would

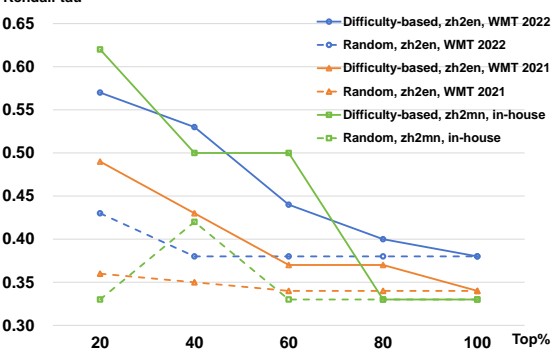

Figure 3: **Three translation evaluation datasets' system ranking correlations with the MQM/DA human evaluations under different subset selection strategies.** Horizontal axis is the top percentage for the selected subset based on the sample's difficulty score ranking. Vertical axis is the Kendall's tau-like correlation coefficient (Freitag et al., 2021b).

also welcome the community to build up support for more languages and more valuable evaluation samples.

# Limitations

It is very costly to do MQM/DA annotation of different systems' performance on *CCEval*, since this must be done by senior linguistic experts mastering both Chinese and our 11 languages. Therefore, we cannot verify the effectiveness of the difficulty-based strategy on the *CCEval* dataset. Rather, we could only validate this hypothesis using one in-house test set with DA annotations of different

systems and two publicly available WMT datasets with rich MQM annotation data.

As a MMT evaluation dataset with diverse distribution, we believe that a larger amount of data samples would increase the evaluation confidence and robustness of *CCEval*. However, due to the cost and time constraints, currently only 2,500 sentence pairs of eleven languages each can be provided, and the coverage in some linguistic dimensions may be still low.

## Acknowledgements

This work was supported by the Major Key Project of Peng Cheng Laboratory, PCL2022D01, the National Key R&D Program of China, 2022ZD0115305, and the National Natural Science Foundation of China, 62106115. We thank all of the translators and linguistic experts, as well as the translation and quality assurance agencies we worked with, for helping create *CCEval*, with special thanks to the Translation Institute of the China International Communications Group.

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

## A  Appendix

### 1.1  *CCEval*'s topic distribution

*CCEval*'s topic distribution in a typical language pair is shown in Table 5.

### 1.2  Translation quality scoring standard

Table 6 shows the translation quality scoring standard used by our linguistic experts during the human quality inspection process.

### 1.3  Details of grammar complexity and word frequency

We employ HanLP[†] for the syntactic analysis of the Chinese texts and count the number of subject-verb-object structures in the text as a measure of grammatical complexity. Regarding word frequency in the text, we first calculate the frequency of all Chinese words in the corpus, and then use the Jieba tokenizer to segment the text. The logarithm of the joint probability of all words is taken as the text's word frequency.

### 1.4  Chinese websites for source collection

We select the Chinese source texts of all evaluation datasets from the following official websites: http://www.news.cn/, https://cn.nytimes.com/, http://www.stdaily.com/, http://fun.lvyou168.cn/, https://www.yaozh.com/

### 1.5  *CCEval*'s data examples

Table 7 demonstrates some data samples of *CCEval* dataset. All of our evaluation samples contain rich metadata information, such as language, domain, topic, etc.

### 1.6  Annotator details

Table 8 shows the detailed information about the translation annotators.

---

[†]https://github.com/hankcs/HanLP

| Topic | #Sentences | Topic | #Sentences |
|---|---|---|---|
| current_affairs | 273 | diet | 24 |
| travel_notes | 238 | emotion | 22 |
| internal_medicine | 110 | gonzo | 22 |
| chat | 107 | agriculture | 22 |
| government_file | 95 | obstetrics_and_gynecology | 20 |
| military | 89 | conference | 28 |
| Chinese_culture | 98 | aerospace | 18 |
| common_sense | 86 | local_news | 17 |
| internationality | 79 | financial_management | 17 |
| physical_education | 66 | mechanical | 16 |
| surgical | 64 | daily_life | 15 |
| culture_and_entertainment | 73 | philosophy | 15 |
| travel_guide | 56 | water_conservancy | 14 |
| life_perception | 54 | environment | 14 |
| popular_science | 46 | dermatology | 14 |
| financial | 55 | health_advisory | 13 |
| travel_exhibition_information | 39 | leader_speech | 13 |
| material | 38 | genome_sequence | 13 |
| ENT | 37 | trip | 12 |
| chemical_industry | 36 | psychiatry | 10 |
| physics | 35 | leader_visit | 10 |
| information_Technology | 35 | shopping | 9 |
| legal | 33 | traditional_Chinese_medicine | 29 |
| pediatrics | 32 | Hong_Kong_Macau_and_Taiwan | 9 |
| life_sciences | 30 | character | 8 |
| biomedical_other_departments | 29 | math | 8 |
| education | 29 | dress | 6 |
| civil_engineering | 28 | drug | 6 |
| industry_analysis | 28 | leader_meet_and_greet | 6 |
| health | 37 | others | 25 |

Table 5: Sentence Count of Each Topic in a Typical Language Pair.

| Score | Description |
|---|---|
| 4 | The translation faithfully reflects source semantics and flows smoothly. |
| 3 | minor errors that will not affect the understanding of the original meaning. |
| 2 | The meaning is generally correct, but there are some localized errors that may cause certain difficulties in understanding. |
| 1 | significant errors that have a severe impact on understanding. |
| 0 | Almost entirely incorrect or completely incomprehensible. |

Table 6: Translation Quality Scoring Standard.

| Chinese source text | Target language | Target text | Domain | Topic |
|---|---|---|---|---|
| 来自加州喷气推进实验室的Robert Carlson表示，一种更好的混合原料是氨和碳氢乙炔。 | Russian | Robert Carlson из калифорнийской Лаборатории реактивного движения сказал, что лучшим сырьём для смеси является аммиак и ацетилен. | Technology | Aerospace |
| 发展改革委、生态环境部有关司局负责同志对清理整改工作提出了要求。 | Portuguese | Os responsáveis relevantes de gabinetes e departamentos, tais como a Comissão Nacional de Desenvolvimento e Reforma e o Ministério da Ecologia e Ambiente, propuseram os requisitos para a limpeza e rectificação. | Technology | Water Conservancy |
| 嘿嘿，路边有一个不大引人注意的广告牌，上面有个女生正专注地看着墙上的招租广告呢。 | Greek | Υπάρχει ένας διαφημιστικός πίνακας στην άκρη του δρόμου που δεν τραβάει την προσοχή πολλών ανθρώπων, στον οποίο ένα κορίτσι παρακολουθεί προσεκτικά τη διαφήμιση ενοικίασης στον τοίχο. | Travel | Travel Notes |
| 建议大家带报纸去坐在店外面吃，别有一番风味。 | Mongolian | Ийм нэгэн онцлогтой тул та бүхнийг сонин авчирч дэлгүүрийн гадаа сууж хооллохыг зөвлөж байна. | Travel | Travel Guide |
| 打扰一下，您能告诉我黄油在哪儿卖吗？ | Turkish | Affedersiniz, tereyağının nerede satıldığını söyleyebilir misiniz? | Daily Life | Shopping |
| 按营业额百分比进行罚款可以提高法律对相关企业违法行为的威慑力。 | Italian | La messa a punto in base alla percentuale di fatturato può aumentare il potere deterrente della legge alle attività illegali delle imprese interessate. | News | Legal |
| 重拳打击涉黑涉恶、电信诈骗、涉枪涉爆、黄赌毒、拐卖妇女儿童等人民群众反映强烈的突出违法犯罪。 | Polish | Uderzanie w gangsterstwo i zło, oszustwa telekomunikacyjne, broń palną i wybuchową, żółte i hazardowe, narkotykowe, handel kobietami i dziećmi oraz inne, podkreślają nielegalne przestępstwa mocno pięścią, co opinia publiczna mocno odzwierciedla. | News | Legal |
| 长时辰运用泻药，可使胃肠道对泻药发作依靠性，除了为解一时之急，最佳仍是少用或不用泻药。 | Czech | Dlouhodobé užívání projímadel může vést k závislosti gastrointestinálního traktu na projímacích záchvatech. Nejlepší je používat projímadla střídmě nebo je nepoužívat vůbec, s výjimkou dočasných naléhavých případů. | Biomedical | Pediatrics |
| 对在保护传承工作中作出突出贡献的组织和个人，按照国家有关规定予以表彰、奖励。 | Arabic | تنظم المؤسسات المعنية المساهمة بقدرة بارزة في العمل بالحماية والوراثة وفقا للوائح الوطنية تقديرات ومكافآت للأفراد والصلة. | Politics | Government File |
| 完善最高人民法院巡回法庭工作机制，健全综合配套措施。 | Croatian | Unaprijediti mehanizam rada okružnog suda Vrhovnog narodnog suda kao i sveobuhvatne mjere podrške. | Politics | Government File |

Table 7: Data Examples of *CCEval*.

| Language | #Annotators | Country of Residence | Age Range | #Years of Professional Translation Experience |
|---|---|---|---|---|
| Mongolian | 6 | Mongolia | 29-42 | 5-16 |
| Lao | 6 | Laos | 29-36 | 5-12 |
| Portuguese | 4 | China & Portugal | 34-44 | 5-9 |
| Polish | 3 | Poland & China | 34-37 | 4-11 |
| Czech | 4 | Czech Republic | 27-54 | 3-21 |
| Italian | 4 | China | 29-42 | 5-17 |
| Russian | 5 | Russia | 32-38 | 7-14 |
| Greek | 3 | Greece | 31-37 | 5-15 |
| Turkish | 4 | Turkey | 33-38 | 4-10 |
| Arabic | 5 | China | 28-35 | 8-13 |
| Croatian | 3 | Croatia | 30-39 | 4-11 |

Table 8: Detailed Information about Translation Annotators.