# OpenReview forum: "CCEval: A Representative Evaluation Benchmark for the Chinese-centric Multilingual Machine Translation"
_EMNLP/2023/Conference — EMNLP 2023 Findings_

### Official Review · Reviewer_Ucq3 · 2023-08-04

**Soundness:** 3

**Excitement:**

3: Ambivalent: It has merits (e.g., it reports state-of-the-art results, the idea is nice), but there are key weaknesses (e.g., it describes incremental work), and it can significantly benefit from another round of revision. However, I won't object to accepting it if my co-reviewers champion it.

**Paper Topic And Main Contributions:**

This paper proposes a new Chinese-centric multilingual machine translation (MMT) evaluation benchmark dataset called CCEval. The main contributions are:

- CCEval is a dataset designed specifically for evaluating Chinese-centric MMT models. It contains 2,500 Chinese source sentences translated into 11 target languages.
- The authors propose methods to select a diverse and representative set of Chinese source sentences based on linguistic features like topic, word frequency, grammar complexity, and translation difficulty.
- The dataset covers a wide range of domains and low/high resource target languages. Quality control measures were taken to ensure high translation quality.
- Experiments show that their proposed translation difficulty metric is effective at selecting sentences that correlate better with human assessments for system ranking.

**Questions For The Authors:**

- It is suggested to validate the effectiveness of this benchmark on actual machine translation models.
- It is recommended to provide some sample data as evidence.
- Please provide more annotation details, such as the number of annotators, educational background, salary, and other relevant information.
- More analysis could be provided characterizing the dataset's diversity and comparability to existing benchmarks. The representativeness for intended use cases could be clarified.

**Reasons To Accept:**

- CCEval fills an important gap as the first open benchmark tailored for Chinese-centric MMT, which is useful.
- The paper provides sensible methods and analysis to construct a high-quality diverse dataset. The difficulty-based sample selection is empirically shown to improve human correlation.
- Releasing the dataset and providing an online evaluation platform is valuable for the research community. The dataset has rich metadata for traceability.

**Reasons To Reject:**

- The scale of the dataset (2,500 sentences) is still quite small for thoroughly evaluating MMT models. Expanding to more target languages and data would be beneficial.
- The author did not actually use this evaluation benchmark to test the machine translation model.
- The empirical verification of the difficulty metric's effectiveness is indirect, performed on different public/private datasets rather than CCEval itself. It lacks sufficient persuasiveness and needs to provide more evidence.
- No data examples were provided.
- Lacks detailed information about data annotation.

**Reproducibility:**

3: Could reproduce the results with some difficulty. The settings of parameters are underspecified or subjectively determined; the training/evaluation data are not widely available.

**Reviewer Confidence:**

3: Pretty sure, but there's a chance I missed something. Although I have a good feel for this area in general, I did not carefully check the paper's details, e.g., the math, experimental design, or novelty.

---

> ### Author Rebuttal · Authors · 2023-08-29
>
> The authors sincerely appreciate your effort and time spent reviewing our paper. Thanks a lot for your constructive comments and encouraging words. They are very helpful in improving the quality of our paper. We have carefully incorporated them in the revised paper. Our point-by-point responses to your comments are given below.
>
> > The scale of the dataset (2,500 sentences) is still quite small for thoroughly evaluating MMT models. Expanding to more target languages and data would be beneficial.
>
> We have thoroughly investigated the current open-sourced evaluation sets, such as FLORES-101 and the evaluation sets of WMT, and we found that the scale of these datasets is mostly in the range of 2000~3000, so we took 2500 as a comparable scale. In addition, our intention of constructing *CCEval* is to hope that as few evaluation samples as possible can be used to achieve reliable and human-like evaluation results. Therefore, we were not keen on expanding the scale of the evaluation set, but rather focused on the overall diversity of the evaluation set and carefully picked each sample, as we described in Section 2. However, we will continue to mine more valuable evaluation samples and supplement them appropriately into *CCEval*.
>
> Regarding the number of included languages, the current paper is only an initial effort of us. In the future, we will continue to expand the dataset and include more languages according to the demands of the Chinese translation evaluation.
>
> > - The author did not actually use this evaluation benchmark to test the machine translation model.
> > - It is suggested to validate the effectiveness of this benchmark on actual machine translation models.
>
> We have actually conducted translation evaluation experiments on our *CCEval* benchmark, using two well-known multilingual translation models, but we were not able to incorporate this section into the current paper due to the page limit of the initial version (4 pages in total). In the revised paper version, given an additional page, we will add this part.
>
> We use SacreBLEU as the evaluation metric, and our evaluation results of SacreBLEU are shown in the table below:
> | **Language** | lo | mn | pt | pl | cs | ru | it | tr | ar | el | hr |
> | ------ | :------: | :------: | :------: | :------: | :------: | :------: | :------: | :------: | :------: | :------: | :------: |
> | **M2M-100, from Chinese** | 1.13 | 0.97 | 18.41 | 13.62 | 13.19 | 17.47 | 17.73 | 15.75 | 15.99 | 16.94 | 13.33 |
> | **M2M-100, to Chinese** | 6.59 | 7.14 | 19.18 | 18.96 | 19.85 | 23.71 | 22.26 | 23.79 | 21.17 | 21.87 | 22.92 |
> | **mBART, from Chinese** | — | 1.03 | 16.31 | 10.24 | 9.98 | 15.45 | 14.21 | 14.21 | 14.22 | 13.22 | 13.01 |
> | **mBART, to Chinese** | — | 8.87 | 15.77 | 13.87 | 17.24 | 21.98 | 18.93 | 18.44 | 19.43 | 19.7 | 19.89 |
>
> The main findings of the evaluation above are:
> - In general, the translation SacreBLEUs from Chinese to other languages are significantly lower than the reverse direction, i.e., from other languages to Chinese;
> - The translation SacreBLEUs between low-resource languages and Chinese are generally far lower than those between high- or medium-resource languages and Chinese.
>
> > The empirical verification of the difficulty metric's effectiveness is indirect, performed on different public/private datasets rather than CCEval itself. It lacks sufficient persuasiveness and needs to provide more evidence.
>
> As we have already mentioned in Sec. 3.1 and the “Limitations” section, there is indeed a lack of human-annotated DA/MQM data to support our effectiveness verification of the difficulty-based sampling method on our *CCEval* evaluation datasets. Currently, we have tried our best and verified it on all available DA/MQM data, including two open-sourced WMT evaluation sets, as well as our own Chinese-to-Mongolian test set.
>
> > - No data examples were provided.
> > - It is recommended to provide some sample data as evidence.
>
> We provide one data sample for each of the 11 language pairs. As shown in the table below, all of our evaluation samples contain rich metadata information, such as language, domain, topic, etc.
> | Chinese source text | Target language | Target text  | Domain | Topic |
> | ------ | :------: | ------ | :------: | :------: |
> | 来自加州喷气推进实验室的Robert Carlson表示，一种更好的混合原料是氨和碳氢乙炔。| ru | "Robert Carlson из калифорнийской Лаборатории реактивного движения сказал, что лучшим сырьём для смеси является аммиак и ацетилен." | Technology | Aerospace |
> | 发展改革委、生态环境部有关司局负责同志对清理整改工作提出了要求。 | pt | "Os responsáveis relevantes de gabinetes e departamentos, tais como a Comissão Nacional de Desenvolvimento e Reforma e o Ministério da Ecologia e Ambiente, propuseram os requisitos para a limpeza e rectificação." | Technology | Water Conservancy |
> | 嘿嘿，路边有一个不大引人注意的广告牌，上面有个女生正专注地看着墙上的招租广告呢。 | el | "Υπάρχει ένας διαφημιστικός πίνακας στην άκρη του δρόμου που δεν τραβάει την προσοχή πολλών ανθρώπων, στον οποίο ένα κορίτσι παρακολουθεί προσεκτικά τη διαφήμιση ενοικίασης στον τοίχο." | Travel | Travel Notes |
> | 建议大家带报纸去坐在店外面吃，别有一番风味。 | mn | Ийм нэгэн онцлогтой тул та бүхнийг сонин авчирч дэлгүүрийн гадаа сууж хооллохыг зөвлөж байна. | Travel | Travel Guide |
> | 打扰一下，您能告诉我黄油在哪儿卖吗？ | tr | "Affedersiniz, tereyağının nerede satıldığını söyleyebilir misiniz? " | Daily Life | Shopping |
> | 按营业额百分比进行罚款可以提高法律对相关企业违法行为的威慑力。 | it | La messa a punto in base alla percentuale di fatturato può aumentare il potere deterrente della legge alle attività illegali delle imprese interessate.  | News | Legal |
> | 重拳打击涉黑涉恶、电信诈骗、涉枪涉爆、黄赌毒、拐卖妇女儿童等人民群众反映强烈的突出违法犯罪。 | pl | "Uderzanie w gangsterstwo i zło, oszustwa telekomunikacyjne, broń palną i wybuchową, żółte i hazardowe, narkotykowe, handel kobietami i dziećmi oraz inne, podkreślają nielegalne przestępstwa mocno pięścią, co opinia publiczna mocno odzwierciedla." | News | Legal |
> | 长时辰运用泻药，可使胃肠道对泻药发作依靠性，除了为解一时之急，最佳仍是少用或不用泻药。 | cs | "Dlouhodobé užívání projímadel může vést k závislosti gastrointestinálního traktu na projímacích záchvatech. Nejlepší je používat projímadla střídmě nebo je nepoužívat vůbec, s výjimkou dočasných naléhavých případů." | Biomedical | Pediatrics |
> | 建议可以将蔬菜和水分一起榨汁饮用，这样既能够摄取蔬菜中的养分，同时又能补充水分。 | lo | "ແນະນຳວ່າຜັກ ແລະ ນ້ຳສາມາດເອົາຄັ້ນນ້ຳມາດື່ມໄດ້, ເຊິ່ງໃຫ້ສານອາຫານໃນຜັກສາມາດດູດຊຶມໄດ້ ແລະ ສາມາດເພີ່ມນ້ຳໄດ້ພ້ອມໆກັນ." | Biomedical | Obstetrics and Gynecology |
> | 对在保护传承工作中作出突出贡献的组织和个人，按照国家有关规定予以表彰、奖励。 | ar | تقديم التقديرات والمكافآة لأفراد أو منظمات التي تقدم مساهمة بارزة في أعمال الحماية والميراث وفقا للوائح الوطنية ذات الصلة. | Politics | Government File |
> | 完善最高人民法院巡回法庭工作机制，健全综合配套措施。 | hr | Unaprijediti mehanizam rada okružnog suda Vrhovnog narodnog suda kao i sveobuhvatne mjere podrške. | Politics | Government File |
>
> > - Lacks detailed information about data annotation.
> > - Please provide more annotation details, such as the number of annotators, educational background, salary, and other relevant information.
>
> We have obtained a complete profile of the annotator details, and will add the required information for each language into the revised paper version. As an example, for each language, we have 3-6 translation annotators.
>
> > More analysis could be provided characterizing the dataset's diversity and comparability to existing benchmarks. The representativeness for intended use cases could be clarified.
>
> We have actually conducted extensive analysis on this, comparing the diversity of our *CCEval* benchmark with that of FLORES-101 dataset from multiple linguistic dimensions, including the distributions of sentence length, word frequency, and grammar complexity. But we were not able to incorporate this part into the current paper due to the page limit of the initial version (4 pages in total). In the revised paper version, given an additional page, we will add this part, including the comparison graph.

---

### Official Review · Reviewer_fSg8 · 2023-08-05

**Typos Grammar Style And Presentation Improvements:** 166
**Soundness:** 3

**Excitement:**

2: Mediocre: This paper makes marginal contributions (vs non-contemporaneous work), so I would rather not see it in the conference.

**Missing References:**

The Flores-200 dataset is one of multiple contributions in the NLLB paper [1]
[1] Costa-jussà, Marta R., et al. "No language left behind: Scaling human-centered machine translation." arXiv preprint arXiv:2207.04672 (2022).

[2] Tiedemann, Jörg. "The Tatoeba Translation Challenge–Realistic Data Sets for Low Resource and Multilingual MT." Proceedings of the Fifth Conference on Machine Translation. 2020.

**Paper Topic And Main Contributions:**

The work presents a machine translation evaluation data set.
The data set is Chinese-centric, paired with 11 other languages: Mongolian, Lao, Portuguese, Polish, Czech, Turkish, Italian, Russian, Arabic, Greek, and Croatian.
The dataset evenly covers 6 domains, and is filtered by hand to ensure quality.
Translations are performed by professional translators, followed by a quality control procedure.
While the resulting dataset is 12-way multiparallel, it can be considered Chinese-centric due to the direction of translation: the source text is always Chinese.

**Questions For The Authors:**

A: Did you evaluate the linguistic diversity of the target sentences? It appears that the claim of high linguistic diversity is based on the diversity-based selection applied to the Chinese source only. If only the Chinese source text is verified to be diverse, then the
claim on line 287 that "The dataset is highly representative and outperforms existing MMT benchmarks in terms of linguistic diversity" must be rephrased to clarify that it applies to Chinese language only.

B: Why do you consider it desirable to transition from English-centric evaluation to another evaluation setting centered around a single language, in this case Chinese? Have you considered the alternative of truly multiparallel evaluation without a central pivot language?

**Reasons To Accept:**

A high-quality machine translation evaluation data set such as this is valuable to the research community focusing on the Chinese language.

**Reasons To Reject:**

The data set covers a fairly small number of languages, only 12.
In particular the Flores-101, Flores-200 [1], and Tatoeba [2] datasets greatly outscale the proposed dataset.

**Reproducibility:**

N/A: Doesn't apply, since the paper does not include empirical results.

**Reviewer Confidence:**

3: Pretty sure, but there's a chance I missed something. Although I have a good feel for this area in general, I did not carefully check the paper's details, e.g., the math, experimental design, or novelty.

---

> ### Author Rebuttal · Authors · 2023-08-29
>
> The authors are very grateful for your constructive comments and suggestions. They are very helpful for us to improve the quality of our paper. We have carefully incorporated them in the revised paper. In the following, your comments are first restated and then followed by our point-by-point responses.
>
> > The data set covers a fairly small number of languages, only 12. In particular the Flores-101, Flores-200 [1], and Tatoeba [2] datasets greatly outscale the proposed dataset.
>
> It is true that our *CCEval* only includes 12 languages, which is not something to show off compared to FLORES-101. However, please note that the initial version of FLORES-101 only contains 2 languages (FLORES-2, [3]), and both FLORES-101 and FLORES-200 are extended versions of FLORES-2. In a similar manner, the current paper is only an initial effort of us. In the future, our *CCEval* will be expanded to include more languages according to the demands from the Chinese translation evaluation.
>
> On the other hand, the Tatoeba corpus is similar to the OPUS corpus, meaning that its portion of evaluation datasets are split from the training datasets, and the evaluation sets have not undergone a strict human annotation and quality control process, just like our *CCEval* or FLORES series evaluation sets have done. Hence, the Tatoeba dataset is not included in our comparison.
>
> Ref: [3] [The FLORES Evaluation Datasets for Low-Resource Machine Translation: Nepali–English and Sinhala–English](https://aclanthology.org/D19-1632) (Guzmán et al., EMNLP-IJCNLP 2019)
>
> > A: Did you evaluate the linguistic diversity of the target sentences? It appears that the claim of high linguistic diversity is based on the diversity-based selection applied to the Chinese source only. If only the Chinese source text is verified to be diverse, then the claim on line 287 that "The dataset is highly representative and outperforms existing MMT benchmarks in terms of linguistic diversity" must be rephrased to clarify that it applies to Chinese language only.
>
> In fact, our target sentences are partially evaluated for diversity, such as domains and topics, which are inherited from the Chinese source. Theoretically, when we decide to choose Chinese as the centric source language, with high-quality translation annotations, the linguistic characteristics of the target languages are determined. In addition, due to the characteristics of different languages, such as the distribution and amount of available corpus, the distribution of Chinese linguistic features may not be completely equivalent to the linguistic feature distribution of the target languages. Therefore, we agree that the usage of “outperforms” in line 287 is inappropriate, since the lead only refers to Chinese and should not include the diversity of other (target) languages. This sentence has been rephrased to “The dataset is highly representative, and the linguistic diversity of Chinese outperforms existing MMT benchmarks.”
>
> > B: Why do you consider it desirable to transition from English-centric evaluation to another evaluation setting centered around a single language, in this case Chinese? Have you considered the alternative of truly multiparallel evaluation without a central pivot language?
>
> Just like FLORES-101 chose English as the centric language, we chose Chinese as the center to construct the multilingual machine translation evaluation set. This is because Chinese is the language with the highest resource level among all of the 12 languages in *CCEval*, thus we can have more optimization flexibility when constructing the diversity of source sentences. In addition, the reason why we cannot use FLORES-101 to evaluate the translation performance between Chinese and other languages is that the sentences of the non-English languages in FLORES-101 are all translationese, hence their text quality might have some problems.
>
> We didn’t consider the circumstance of multi-parallel evaluation without a central language, for two reasons: a) the translation annotation cost of building a multi-parallel dataset is on the order of $n^2$ versus the cost of *n* in our case, where *n* is the number of total languages. b) It is very difficult, if not impossible, to pick and translate enough sentences for the language pairs between two low-resource languages.
>
> > Missing References: [1] FLORES-200 dataset; [2] Tatoeba dataset
>
> We have already cited FLORES-200 [1] and will include Tatoeba [2] in the revised paper version.
>
> > 166: "Difficulty" should be formatted as text, rather than a sequence of variables. Currently the text is italic and the kerning is not correct.
>
> Thanks for pointing out this mistake. We will correct it in the revised paper version.
>
> > Table 3: Please add the number of sentence pairs in each data set. To fit the additional column, you can either abbreviate or rotate the column headings.
>
> Sure. We will add this additional column/row in the revised paper. Below please find the number of sentence pairs for each dataset listed in Table 3:
> | **Dataset** | FLORES-101/200 | ALT | TICO-19 | CCEval |
> | ------ | :------: | :------: | :------: | :------: |
> | **#Sentence pairs** | 3001 | 20000 | 3071 | 2500 |

---

### Official Review · Reviewer_DEN2 · 2023-08-11

**Soundness:** 3

**Excitement:**

3: Ambivalent: It has merits (e.g., it reports state-of-the-art results, the idea is nice), but there are key weaknesses (e.g., it describes incremental work), and it can significantly benefit from another round of revision. However, I won't object to accepting it if my co-reviewers champion it.

**Paper Topic And Main Contributions:**

This paper presents a Chinese-centric multilingual machine translation evaluation dataset. The dataset consisting 11 languages which covers 6 domains and 67 topics.

**Questions For The Authors:**

1. I would like to know the average number of words/tokens per sentence.

**Reasons To Accept:**

The authors proposed a Chinese centric multilingual machine translation benchmark evaluation dataset consisting 2500 Chinese sentences. The authors also conducted experiments on the evaluation dataset to compare with existing multilingual datasets.

**Reasons To Reject:**

1. In L101, the authors employed various linguistic features but didn’t discuss which linguistic features were used in this study.
2. In Equation 1, I didn’t really understand the relation between the source sentence and the restored source sentence by the BERT model. The similarity of these two sentences really matters for the translation of a sentence by humans? In my observance, difficulties in translation might happen for long or complex sentences, because the translator lost the context sometimes.
3. Why were long sentences over 60 characters removed?
4. LSP’s translation can be modified while verifying the quality of the translation. Why the authors didn’t fix the translation or authors keep them on purpose is not discussed properly.
5. 8 topics have 1 to 3 individual sentences, would it be insightful to draw a conclusion based on 1 to 3 translations for a topic?
6. 2,500 sentences are not enough to evaluate an MMT model.

**Reproducibility:**

N/A: Doesn't apply, since the paper does not include empirical results.

**Reviewer Confidence:**

3: Pretty sure, but there's a chance I missed something. Although I have a good feel for this area in general, I did not carefully check the paper's details, e.g., the math, experimental design, or novelty.

**Typos Grammar Style And Presentation Improvements:**

There are many grammatical errors in the article. The authors should use a grammatical tool for correction.

---

> ### Author Rebuttal · Authors · 2023-08-29
>
> The authors are very grateful for your constructive comments and suggestions. They are very helpful for us to improve the quality of our paper. We have carefully incorporated them in the revised paper. In the following, your comments are first restated and then followed by our point-by-point responses.
>
> > In L101, the authors employed various linguistic features but didn’t discuss which linguistic features were used in this study.
>
> We mentioned and discussed the specific linguistic features to use in the “Diversity-based selection” section of Sec. 2.1, i.e., domain, topic, word frequency, and grammar complexity.
>
> > In Equation 1, I didn’t really understand the relation between the source sentence and the restored source sentence by the BERT model. The similarity of these two sentences really matters for the translation of a sentence by humans? In my observance, difficulties in translation might happen for long or complex sentences, because the translator lost the context sometimes.
>
> Intuitively, long or complex sentences are indeed difficult to translate, since humans need to focus on more linguistic aspects during translations. However, there is currently no quantitative metric to measure the degree of "difficulty" in terms of sentence length or complexity. Therefore, in light of the sentence translation characteristics disclosable by back translations, we borrow the idea of denoising autoencoder (DAE) to characterize the degree of translation difficulty of a sentence.
>
> > Why were long sentences over 60 characters removed?
>
> Three reasons:
> 1. In Chinese, the proportion of sentences longer than 60 characters is very small (no more than 5%).
> 2. The translation annotation of super long sentences is difficult and costly.
> 3. In our early pilot assessments of translation quality, the quality scores of these super-long sample sentences were too low.
>
> > LSP’s translation can be modified while verifying the quality of the translation. Why the authors didn’t fix the translation or authors keep them on purpose is not discussed properly.
>
> There is not a 100% accurate dataset, since "professional" is always relative, meaning that theoretically we can always find more professional language experts to discover translation mistakes that are not able to be revealed by less professional experts, hence there is no absolute 100% for the quality score. Therefore, our goal for the dataset construction can only be set as achieving as high as possible quality scores given by highly qualified professionals. In fact, from the experience of previous datasets inspected by our group of linguistic experts, \>90% quality score given by these experts does indicate a very high-quality dataset.
>
> > 8 topics have 1 to 3 individual sentences, would it be insightful to draw a conclusion based on 1 to 3 translations for a topic?
>
> It is true that some topics have too few sentences, but our dataset does not intend to compare the evaluation results of different topics. In addition, in the revised paper, we will merge the longtail low-frequency topics into a single category, and will name it as the "other" category.
>
> > 2,500 sentences are not enough to evaluate an MMT model.
>
> When constructing an evaluation dataset, its scale is the first issue that needs to be considered. We have thoroughly investigated the current open-sourced evaluation sets, such as FLORES-101 and the evaluation sets of WMT, and we found that the scale of these datasets is mostly in the range of 2000~3000, so we took 2500 as a comparable scale. In addition, our intention of constructing *CCEval* is to hope that as few evaluation samples as possible can be used to achieve reliable and human-like evaluation results. Therefore, we were not keen on expanding the scale of the evaluation set, but rather focused on the overall diversity of the evaluation set and carefully picked each sample, as we described in Section 2. However, we will continue to mine more valuable evaluation samples and supplement them appropriately into *CCEval*.
>
> > I would like to know the average number of words/tokens per sentence.
>
> | **Language** | zh | lo | mn | pt | pl | cs | ru | it | tr | ar | el | hr |
> | ------ | ------ | ------ | ------ | ------ | ------ | ------ | ------ | ------ | ------ | ------ | ------ | ------ |
> | **Mean #tokens/sentence** | 30 | 36 | 31 | 27 | 33 | 31 | 33 | 29 | 26 | 35 | 29 | 28 |
>
> > There are many grammatical errors in the article. The authors should use a grammatical tool for correction.
>
> Thank you very much for the suggestion. We have used “Grammarly” to check over the full paper carefully, and have corrected all of the found mistakes. Here are three examples:
> - Line 031: “the most population” has been corrected to “the largest population”
> - Line 187: “work flow” has been combined to “workflow”
> - Line 245: “as regard to” has been corrected to “in regard to”

---

### Meta-Review · Area_Chair_7cD9 · 2023-09-15

**Recommendation:** 3

**Metareview:**

This paper work presents a Chinese-centric machine translation evaluation dataset paired with 11 other languages. It covers 6 domains and is filtered by hand to ensure quality. This is a valuable resource since most other evaluation datasets are either not curated or not Chinese-centric, but it remains unclear if the proposed resource provides any practical advantages over other evaluation datasets like Flores or Tatoeba.

During the author response period, the authors showcased the effectiveness of their benchmark on actual machine translation models, and they also promised to provide a comparison between their dataset and Flores, which hopefully addresses some criticisms made by the reviewers.

---

### Decision · Program_Chairs · 2023-10-07

**Decision:**

Accept-Findings

**Comment:**

This paper work presents a Chinese-centric machine translation evaluation dataset paired with 11 other languages. It covers 6 domains and is filtered by hand to ensure quality. This is a valuable resource since most other evaluation datasets are either not curated or not Chinese-centric, but it remains unclear if the proposed resource provides any practical advantages over other evaluation datasets like Flores or Tatoeba.

During the author response period, the authors showcased the effectiveness of their benchmark on actual machine translation models, and they also promised to provide a comparison between their dataset and Flores, which hopefully addresses some criticisms made by the reviewers.